# Excessive Fructose Intake Impairs Baroreflex Sensitivity and Led to Elevated Blood Pressure in Rats

**DOI:** 10.3390/nu11112581

**Published:** 2019-10-25

**Authors:** Hsin-Hung Chen, Chih-Hsun Chu, Shu-Wei Wen, Chi-Cheng Lai, Pei-Wen Cheng, Ching-Jiunn Tseng

**Affiliations:** 1Department of Medical Education and Research, Kaohsiung Veterans General Hospital, Kaohsiung 813, Taiwan, shchen0910@gmail.com (H.-H.C.); hhchen0910@vghks.gov.tw (S.-W.W.); 2Yuh-Ing Junior College of Health Care & Management, Kaohsiung 821, Taiwan; 3Division of Endocrinology and Metabolism, Department of Internal Medicine, Kaohsiung Veterans General Hospital, Kaohsiung 813, Taiwan, chchu@vghks.gov.tw; 4Department of Nursing, Fooyin University, Kaohsiung 831, Taiwan; 5Cardiology, Kaohsiung Municipal United Hospital, Kaohsiung 804, Taiwan, llccheng@gmail.com; 6Institute of Biomedical Sciences, National Sun Yat-sen University, Kaohsiung 804, Taiwan; 7Department of Pharmacology, National Defense Medical Center, Taipei 114, Taiwan; 8Department of Medical Research, China Medical University Hospital, China Medical University, Taichung 404, Taiwan

**Keywords:** baroreflex sensitivity, blood pressure, fructose, nucleus tractus solitarii, sympathetic nerve activity, nitric oxide

## Abstract

Hypertension development with an increased intake of added sugar, especially excessive fructose intake, was shown in the National Health and Nutrition Examination Survey (NHANES) data. However, the mechanism underlying blood pressure (BP) elevation with increased fructose intake is still unclear. First, the present study showed that in rats fed 10% fructose for one week, BP and fructose/glucose levels increased in the central and peripheral nervous system. Furthermore, increased fructose intake resulted in an upregulation of fructose concentration in the cerebrospinal fluid. Second, consumption of excess fructose increased serum triglycerides. However, the inhibition of triglyceride production did not mitigate sympathetic nerve hyperactivity, but contributed to an insignificant decrease in BP. Finally, increased fructose intake reduced nitric oxide (NO) levels in the nucleus tractus solitarii (NTS) and reduced baroreflex sensitivity within a week. Collectively, the data suggested that fructose intake reduced NO levels in the NTS and caused baroreflex dysfunction, which further stimulated sympathetic nerve activity and induced the development of high BP.

## 1. Introduction

Hypertension is the most common chronic disease in developed countries [1]. Numerous studies have shown that the prevalence of hypertension in the United State has increased from approximately 5% to 10% of the adult population to approximately 31% since the early 20th century [2]. There has been a significant ∼10% to 20% increase in mean total fructose intake in children, and a marked ∼20% to 60% increase in teenagers and adults between 1978 and 2004 [3,4]. Obesity, type 2 diabetes, and cardiovascular diseases were shown to be linked to fructose consumption in large amounts [5,6]. High fructose consumption has shown greater adverse effects on metabolism and vascular health than glucose consumption in several animal studies, and it causes detrimental health effects, such as fatty liver diseases [7]. In addition, a high-fructose diet (60%) and intake of 10% fructose water over an eight-week period induced hypertension, hypertriglyceridemia, and hyperuricemia in rats [8]. Fructose, a sweeter alternative to glucose, is usually used in smaller amounts than glucose. Because fructose cannot be used directly as a source of energy, the metabolisms of fructose and glucose are very different [9]. The actual mechanism of how fructose affects the central nervous system (CNS) and subsequently influences BP regulation is still unknown. 

Many studies indicated that the mechanism of BP elevation due to excessive fructose consumption falls into three broad categories: chronic stimulation of the sympathetic nervous system (SNS), dysfunction of the endothelium, and upregulation of salt absorption [10]. There is evidence suggesting that the chronic sympathetic nerve is overactivated in fructose-induced hypertension. Fructose-fed rats have significantly higher serum norepinephrine and triglycerides levels than the control rats, and these parameters are involved in increases in BP [11]. Previous studies revealed that chemical sympathectomy restored increased BP in fructose-fed rats, suggesting that the development of hypertension is linked to the function of the SNS [12]. Information in the sympathetic and parasympathetic nerves is transmitted by renal afferent fibers. The information converges at the nucleus tractus solitarii (NTS), which is the primary site of BP and sympathetic nerve activity (SNA) modulation [13,14]. Experimental lesions in the NTS cause a loss of baroreflex control of BP and sympathetic activation, and evoke severe hypertension in animals [15,16]. In this study, we focused on the mechanisms underlying BP elevation after excessive fructose intake for one week. In a previous study, compared with those in glucose-fed rats, serum triglyceride levels increased to a greater degree in fructose-fed rats [17]. Our previous studies revealed that serum triglyceride concentrations increased much more significantly in the group fed 10% fructose water than in the control group [18]. Therefore, we speculate that increased serum triglyceride levels caused by increased fructose intake may be a possible trigger for elevated BP. Moreover, we examined whether excessive consumption of fructose for one week is associated with sympathetic nerve overactivation and impairment of baroreflex sensitivity.

Epidemiologic studies all pointed to the fact that fructose may be involved in hypertension; however, there is no direct evidence to support such a claim. A previous study showed that fructose acts as a stimulus for triglyceride synthesis, triggering sympathetic nerve overactivation in the NTS following fructose intake for four weeks [18]. In this study, we hypothesized that increased fructose intake decreased nitric oxide (NO) level in the NTS, diminished baroreflex sensitivity, increased serum triglycerides, and further overactivated SNS, thereby increasing BP. This study aimed to investigate whether increased fructose intake induces BP elevation and to determine the mechanisms in rats fed fructose for one week. Here, we show that a one-week consumption of 10% fructose water upregulated BP, leading to hyperactivation of the SNS and impairment of baroreflex sensitivity. 

## 2. Materials and Methods 

### 2.1. Animal Care and Experiments

Eight-week-old male Wistar Kyoto (WKY) rats were obtained from the National Science Council Animal Facility (Taipei, Taiwan) and housed in the animal room of Kaohsiung Veterans General Hospital (Kaohsiung, Taiwan). The rats were kept in individual cages in a room with controlled lighting (12-h light/12-h dark cycle) and maintained temperature at 23 ℃ to 24 ℃. The rats were provided with normal rat chow (Purina, St. Louis, MO, USA) and tap water *ad libitum*. All animal research protocols had been approved by the Research Animal Facility Committee of Kaohsiung Veterans General Hospital (VGHKS-104-A011) and the use of Laboratory Animals published by US National Institutes of Health [19]. 

Before the study, all rats were acclimatized to the housing conditions for one week. Next, the animals were trained to accustom them to the indirect BP measurement procedure for one week. The rats were randomly divided into two groups as follows: (1) Control group, WKY rats were provided drinking water *ad libitum* for 0 to 7 days (*n* = 6 each); (2) Fructose group, WKY rats were provided with 10% fructose water ad libitum for 0 to 7 days (*n* = 5–7 each). The two groups were continuously measured for body weight, diet, drinking water volume (water intake), urine volume, and systolic blood pressure (SBP) every day (Figure 1). The rats were placed individually in metabolic cages (one per cage) (Braintree Scientific, Inc., Braintree, MA, USA). Their diet and drinking water were measured, and urine collected after 24-h at 10:00 a.m. each day. A measuring bottle was used to measure daily drinking water volume and urine volume.

We further measured the levels of serum biochemical parameters, renal SNA, baroreflex response sensitivity (BRS), and NO in the NTS of the fructose groups for 0 to 7 days (Figure 1). In addition, we used PF04620110, a diacylglycerol acyltransferase 1 (DGAT1) inhibitor [20] that catalyzes the final committed step in triglyceride biosynthesis. To confirm whether fructose-induced hypertension was inhibited by triglyceride inhibition, rats were divided into four groups in this one-week experiment: (1) Control (WKY rats fed drinking water for a week, *n* = 6); (2) Control+PF04620110 (WKY rats orally administered PF04620110 0.1 mg/kg/day for a week, *n* = 5–6); (3) Fructose (WKY rats fed 10% fructose water for a week, *n* = 4–8); and (4) Fructose+PF04620110 (WKY rats fed 10% fructose water and orally administered PF04620110 for a week, *n* = 4–8) groups.

### 2.2. Blood Pressure Measurement

A CODA 8 noninvasive BP monitoring system for rats (Kent Scientific, Torrington, CT, USA) was used for all tail-cuff measurements every day. This system uses a volume-pressure recording (VPR) to detect BP based on volume changes in the tail, as previously described [21]. The CODA system was factory-calibrated and used with standard settings and recommendations [22] as follows. All rats were acclimated to the restrainer for 10–20 min per day for at least three days before the start of the experiment. Thereafter, the rats were subjected to experimental protocols as detailed below. The rats were warmed for 5 min before and during BP recording. To measure BP, the occlusion cuff was inflated to 250 mm Hg and deflated over 15 s. The VPR sensor cuff detected changes in the tail volume as the blood returns to the tail during the occlusion cuff deflation. Each recording session consisted of 20 to 30 inflation and deflation cycles per set, of which the first five cycles were “acclimation” cycles and not used in the analysis, whereas the subsequent cycles were used. A number of 20 to 30 individual readings were obtained in rapid succession during the measurement. The highest and lowest readings were eliminated, and 10 to 20 remaining readings were averaged. VPR technology is applicable for high-throughput experiments, as recommended by the American Heart Association, and this technology has been validated [23]. 

### 2.3. Determination of Nitric Oxide Analysis in the NTS 

The NTS (15 mg) were deproteinized by a microcon YM-30 filter (Millipore, Bedford, MA, USA). The total amount of NO in the NTS was determined by a Sievers Nitric Oxide Analyzer purge system (NOA 280i; Sievers Instruments, Boulder, CO, USA) using the chemiluminescence-based procedure defined by the manufacturer, with modification. The NO level was measured by ozone-induced chemiluminescence and corrected for that in the NTS of both the control and 10% fructose groups, as previously described [18]. The measurement for each sample was conducted in triplicate. 

### 2.4. Renal Sympathetic Nerve Activity Recording

The renal nerve in anesthetized rats was identified through the retroperitoneal incision with the assistance of an operating microscope (Zoletil50, 50 mg/kg, intraperitoneal). The distally cutting renal nerve was placed on a pair of silver recording electrodes (only one-way signals were recorded to avoid two-way signal interference) and immersed in warm mineral oil (for insulation). The signals were amplified, passed through a band pass filter and displayed on an oscilloscope, then renal SNA was recorded as previously described [14]. 

### 2.5. Baroreflex Sensitivity Recording

Under anesthesia with Zoletil50 (50 mg/kg intraperitoneal, and 10 mg/kg intravenous if necessary) in rats, a polyethylene cannula (PE 50) was inserted into their abdominal aorta through the right femoral artery to measure the mean arterial pressure and heart rate. A second cannula was inserted into the femoral vein for phenylephrine administration. The baroreflex response sensitivity (BRS) [24] of the control and fructose groups under anesthesia were elicited by increasing doses of phenylephrine (10 to 30 μg/kg, i.v.) to test phenylephrine-induced BRS in the rats. 

### 2.6. Assay of Serum Biochemical Parameters 

Heart blood and cerebrospinal fluid (CSF) samples were collected after the rats were sacrificed with excess CO2. Fasting fructose levels in serum and CSF (EFRU-100; EnzyChrom Fructose assay kit) as well as fasting glucose levels in CSF (EBGL-100; EnzyChrom glucose assay kit) were measured using an assay kit (BioAssay System, Hayward, CA, USA) and detected using a Biochrom Anthos Zenyth 200rt Microplate Reader (Cambridge, UK). Levels of serum total cholesterol, direct low-density lipoprotein (LDL), direct high-density lipoprotein (HDL), triglycerides, and glucose were determined using a Clinical Chemistry Analyzer (Ortho Clinical VITROS™ 350 System, Raritan, NJ, USA). 

### 2.7. Statistical Analysis

All data are expressed as means and standard error of the mean (SEM). The Mann–Whitney *U*-test (for comparison of the control and 10% fructose groups) or the Kruskal–Wallis test followed by a one-way analysis of variance was applied to compare differences between groups. *P* < 0.05 was considered significant. Post-hoc power analysis using G-Power 3.1.9.2 software was used to determine the power of the present study. Based on the results for all other variables, effect sizes from 0.2 to 14.6 were calculated. With these effect sizes, alpha of 0.05, and sample size from 4 to 8, a power ranging from 0.06 to 0.99 was calculated.

## 3. Results

### 3.1. Fructose Intake for One Week Significantly Increases Blood Pressure in Rats 

We examined whether the intake of 10% fructose water for one week leads to increased BP. We investigated BP changes in fructose-fed rats from day 0 to day 7, and measured fructose and glucose concentrations in the central and peripheral compartments. Initially, water intake showed no difference among the groups (control group, 34.5 ± 0.7 mL vs. fructose group, 32.0 ± 1.6 mL, Figure 2A). However, after an intake of 10% fructose water, water intake significantly increased (33.3 ± 1.1 mL vs. 49.8 ± 3 mL). After consuming fructose water for two days, the water intake in the rats peaked (32.3 ± 1.1 mL vs. 71.8 ± 4.5 mL). Furthermore, after seven days, the volume of water intake in the 10% fructose group was two times higher compared with the control group (33.8 ± 1.7 mL vs. 66.8 ± 2.4 mL). Interestingly, we observed a continued increase in SBP during seven days of 10% fructose water intake (107 ± 1.1 mmHg to 137.5 ± 2.7 mmHg, Figure 2B). We speculated that the increase in SBP is related to fructose intake. Therefore, we measured fructose and glucose levels in rat serum and CSF. On day 1, compared to that in the control group, fructose levels in CSF increased (1.35 ± 0.13 mg/dL vs. 2.04 ± 0.06 mg/dL, Figure 2D), but serum fructose levels (Figure 2C) did not significantly increased. During the seven-day experiment, fructose and glucose levels in serum and CSF continued to increase (Figure 2C–F). The increase in SBP may be positively correlated with fructose water consumption. In this seven-day experiment, body weight and diet did not change much, but drinking water and urine volumes fluctuated throughout the study period (Appendix A). In addition, we examined metabolism of serum lipids, such as total cholesterol, LDL cholesterol, HDL cholesterol, and triglyceride levels, for seven days. We discovered that the triglyceride level increased significantly in the serum of 10% fructose water-fed rats (Appendix A). 

### 3.2. Inhibition of Triglyceride does not Decrease Sympathetic-Nerve Activity after 1 Week of Fructose Intake

If the BP of rats increased after consumption of 10% fructose water, we speculated that the increase in BP is caused by triglycerides. Therefore, we investigated whether inhibition of triglyceride production (using the DGAT1 inhibitor PF04620110) reduces BP in rats fed fructose for seven days. In this study, oral administration of PF04620110 led to a slight decrease in the SBP of rats on day 7 (137.5 ± 2.7 mmHg vs. 129.1 ± 1.7 mmHg, Figure 3A). The oral administration of PF04620110 significantly decreased serum triglyceride levels in fructose-fed rats (180.3 ± 11.8 mg/dL vs. 129.0 ± 15.7 mg/dL, Figure 3B), but did not alter SNA in fructose-fed rats on day 7 (Figure 3C,D). We further confirmed that the oral administration of PF04620110 did not elevate NO levels in the NTS of fructose-fed rats on day 7 (Figure 3E). 

### 3.3. Fructose Intake Induces Sympathetic Hyperactivity and Decreases Nitric Oxide Level in Rats Fed Fructose for One Week

The next experiment determined whether intake of 10% fructose water for one week increases SBP and BP. We investigated SNA and NO levels in the NTS of fructose-fed rats for seven days. The time course of SNA during the experiment is shown in Figure 4A. On day 1, SNA in fructose-fed rats increased significantly (55.6 ± 9.6 bursts/min vs. 85.3 ± 9.2 bursts/min, Figure 4B), compared to that on day 0, and SNA continued to increase until day 7 (111.4 ± 16.1 bursts/min). We further observed that NO level in the NTS of fructose-fed rats also decreased (3.8 ± 0.4 μM/μg vs. 2.6 ± 0.2 μM/μg), compared to that on day 0, and NO level continued to decline until day 7 (1.4 ± 0.3 μM/μg, Figure 4C). Moreover, the data showed that RSNA and nitrate levels in the NTS did not change on the 0th, 3rd, and 7th days compared with the control of rats by drinking water (Appendix A). 

### 3.4. Fructose Intake for One Week Impairs Baroreflex Response Sensitivity in Rats

We further investigated whether intake of 10% fructose water impairs BP regulation by the CNS. Dose-dependent of phenylephrine (10 to 30 μg/kg) was intravenously injected throughout a seven-day period to increase the BP and pulse period in fructose-fed rats (Figure 5A). In fructose-fed rats, BRS significantly decreased (3.0 ± 0.9 ms/mmHg vs. 1.2 ± 0.2 ms/mmHg) at day 1, compared with that at day 0, and continued to decrease for seven days (0.1 ± 0.2 ms/mmHg, Figure 5C). Moreover, the data showed that BRS did not change on different days in the control of rats by drinking water on the 0th, 3rd, and 7th days, respectively (Appendix A). BRS only in fructose-fed rats decreased significantly over seven days.

## 4. Discussion

Here, we describe a link between fructose and NO levels in the NTS, as well as its association with baroreflex sensitivity and activation of the CNS. Our findings showed that excessive fructose consumption increased BP and renal SNA, decreased NO bioavailability, and impaired baroreflex response. Excessive fructose intake increased serum triglycerides levels and induced BP elevation by stimulating SNA; however, alleviation of hypertriglyceridemia did not attenuate sympathetic activation and only slightly lowered BP. It was suggested that, mechanistically, SNA played a role in modulating changes in BP due to increased fructose consumption through another unknown pathway. 

Epidemiological studies have hinted at a link between fructose consumption and elevated blood pressure. Jalal et al. [25] reported that excess dietary fructose (≥74 g/day) in the form of added sugar was associated with higher BP values in adults in the United States who did not have a history of hypertension. Similarly, a study of 4867 adolescents found that SBP rose by 2 mmHg following the intake of sugar-sweetened beverages from the lowest to the highest category [26]. In a prospective study involving adults in the United States, Chen et al. [27] found that intake of one less sugar-sweetened beverage per day was associated with a 1.8 mmHg reduction in SBP and a 1.1 mmHg reduction in DBP over 18 months. Some animal studies showed that a high-fructose diet was associated with hypertriglyceridemia, hyperinsulinemia, impaired glucose tolerance, insulin resistance, and increased BP and body weight [28]. However, these adverse effects were not indicated with equivalent calories of glucose. In this study, increased fructose intake for 0 to 2 days evoked increases in BP levels, and the high BP levels were maintained until day 7 (Figure 2). 

The great increase in fructose consumption in the last decades leads to the rapid accumulation of triglyceride in the Occidental population [29]. Tran et al. reported [11] that serum norepinephrine and triglyceride levels are significantly increased in fructose-fed hypertensive rats, as reflected by high SNA. According to Tran et al., hypertension can develop in rats owing to excessive fructose consumption, and the developed hypertension can be treated using a sympatholytic agent called prazosin, which blocks the α1-adrenoreceptors [30]. Hypertension inhibition by prazosin did not reduce the level of triglyceride in serum. Consistent with this previous result, our data showed that the use of a DGAT1 inhibitor for inhibiting serum triglyceride level led to a slight decrease in BP but did not affect SNA (Figure 3). 

The baroreflex system is an important mechanism in the regulation of heart rate, sympathetic tone, and consequently BP. Impairment of baroreflex sensitivity is involved in the sympathoexcitatory and sympathoinhibitory effects of metabolic syndromes [31]. A previous study showed that hypertriglyceridemia may be the main contributory factor associated with impaired baroreflex sensitivity during a metabolic syndrome [32]. Soncrant et al. reported that elevated BP was increased by sympathoexcitation in 20% fructose-induced, salt-sensitive, hypertensive rats [33]. It has been shown that the ingestion of fructose altered the secretion of hormones that regulate energy balance associated with increased SNA [34]. However, in the current study, although a DGAT1 inhibitor was administered orally to block triglycerides production, sympathoexcitation was not inhibited in fructose-fed rats. Dos Santos et al indicated that fructose consumption similarly reduced both baroreflex sensitivity and activity [35]. Our research further indicated that fructose intake may reduce NO levels in the NTS and cause baroreflex dysfunction, which further stimulates SNA and induces the development of high BP (Figure 4 and Figure 5). 

Generally, anesthetics affect the basal levels of sympathetic nerve activity to regulate blood pressure, which plays an important role in controlling cardiovascular function [36,37]. Sun et al. indicated that anesthesia attenuated the excitatory response of RSNA and HR to anaphylactic hypotension, while these excitatory responses were attenuated by anesthetics in the order ketamine-xylazine > urethane = pentobarbital [38]. Bencze et al. found that pentobarbital anesthesia had a modest influence on the BP level and its maintenance by the above vasoactive systems [37]. In addition, anesthetics may inhibit the primary area in the baroreceptor reflex pathway of the central nervous system, resulting in attenuation of the baroreceptor reflex [39]. Therefore, sympathetic nerve activity and cardiovascular function may be affected by the anesthetic. However, there were some limitations to our study. First, we measured BP during consciousness, whereas RSNA and baroreflex sensitivity measurements were performed under the anesthetic. Second, our sample size was relatively small and the study used nonparametric methods for analysis. Third, the fructose group was used as the control group only on day 0.

With regards to the relationship between baroreflex sensitivity and NO level in the NTS, our previous studies indicated that central endogenous NO is involved in the medullary regulation of BP and that a NO synthase inhibitor attenuates baroreflex activation [40]. In addition, unilateral microinjection of the nitric oxide synthase (NOS) inhibitor L-NMMA into the NTS generates dose-dependent bradycardic effects, but these effects were inhibited after intravenous injection of atropine, indicating that the bradycardic effect of L-NMMA is mediated by baroreflex responses [41]. In terms of the mechanisms by which fructose in the CSF induces suppression of NO levels in the NTS and further damages baroreflex sensitivity, our previous studies indicated that superoxide production increases in the NTS of rats fed fructose for one week [42]. Fructose-induced neurogenic hypertension might be mediated by the activation of p38, followed by phosphorylation of the insulin receptor substrate 1 ser307, which might occur via superoxide overexpression in the NTS [43]. In this study, we showed that fructose intake may reduce NO levels in the NTS and cause baroreflex dysfunction, which further stimulated SNA and induced the development of high BP. 

## 5. Conclusions

In conclusion, this study elucidated the underlying mechanisms of fructose-induced BP elevation in a one-week experiment. The results of the study indicated that excessive fructose intake increased fructose concentrations in the CSF and led to the development of high BP caused by sympathetic hyperactivity, suggesting that the CNS played a critical role in elevating BP through reduction of NO in the NTS, as well as impaired baroreflex sensitivity. 

## Figures and Tables

**Figure 1 nutrients-11-02581-f001:**
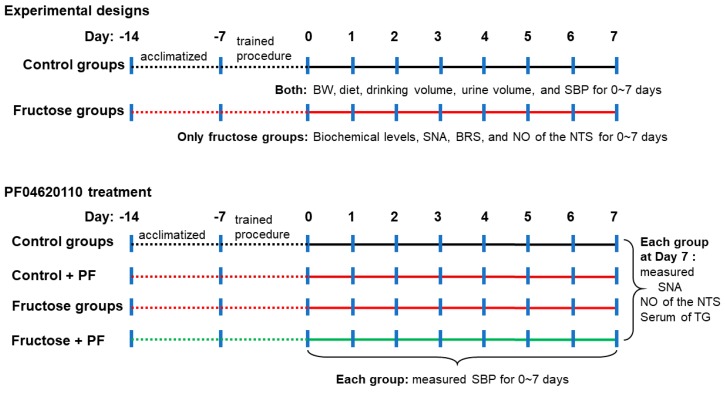
**Experimental design.** BW, body weight; SBP, systolic BP; SNA, sympathetic nerve activity; BRS, baroreflex response sensitivity, NO, nitric oxide; NTS, nucleus tractus solitarii; TG, triglyceride; PF, PF04620110 (DGAT1 inhibitor).

**Figure 2 nutrients-11-02581-f002:**
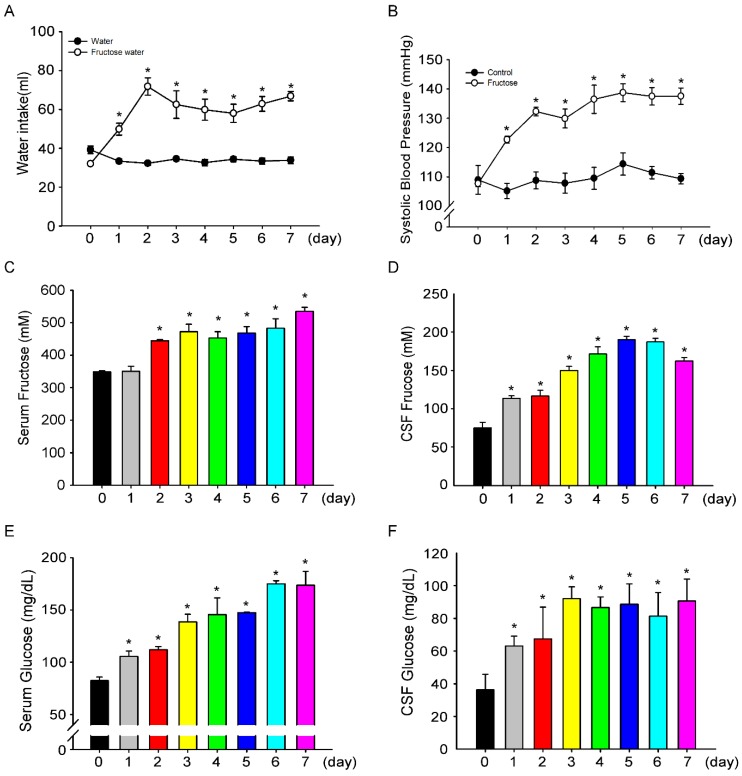
**Intake of 10% fructose water increases SBP and fructose concentrations in CSF and serum.** (**A**) Daily water intake recorded for seven days. The filled circles (●) represent the control group, open circles (○) represent the fructose group. (**B**) Time course of SBP examined in each group for seven days. (**C**) Fructose levels in the serum of fructose-fed rats were determined for seven days. Fructose level in the serum of fructose-fed rats significantly increased from day 2 to day 7, compared to that on day 0. (**D**) Fructose levels in the CSF of fructose-fed rats were determined for seven days. The fructose level of in the CSF of fructose-fed rats significantly increased from day 1 to day 7, compared to that on day 0. (**E**) Glucose levels were determined in the serum of fructose-fed rats for seven days. The glucose level in the serum of fructose-fed rats significantly increased from day 1 to day 7, compared to that on day 0. (**F**) Glucose levels in the CSF of the fructose-fed rats were determined for seven days. The glucose level in the CSF of the fructose-fed rats significantly increased from day 1 to day 7, compared to that on day 0. The data are presented as the mean ± SEM (*n* = 4–8). * *p* < 0.05 versus the control group and fructose group day 0.

**Figure 3 nutrients-11-02581-f003:**
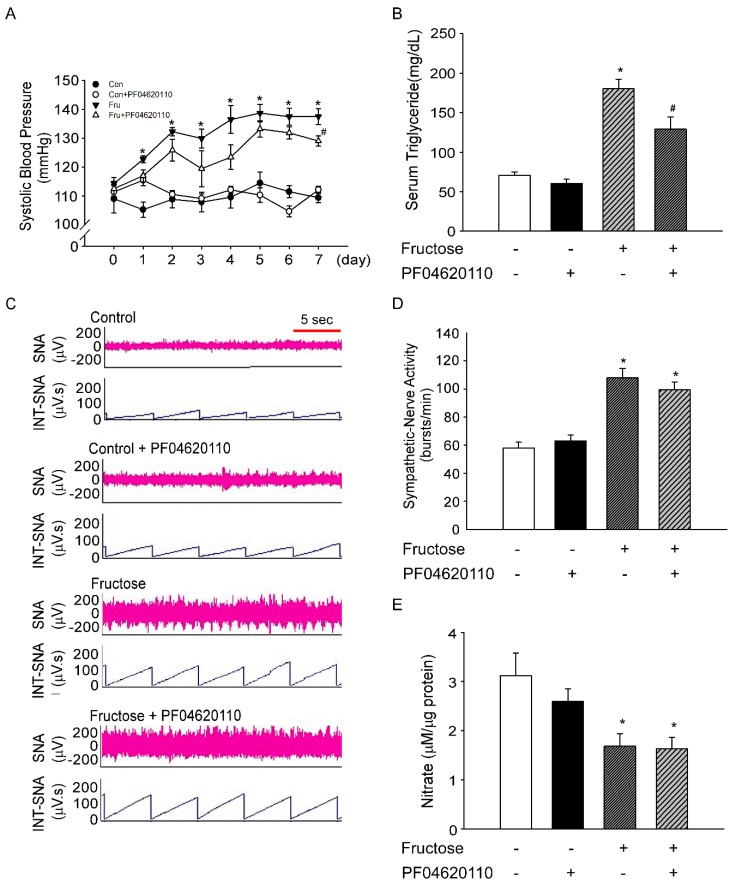
PF04620110 decreases SBP and triglyceride levels but does not affect SNA in rats fed 10% fructose water. (**A**) Time course of SBP. SBP was examined after oral administration of PF04620110 for seven days. The filled circles (●) represent the control group, and open circles (○) represent the control+PF04620110 group. The inverted filled triangles (▼) represent the fructose group, and open triangles (△) represent the fructose+PF04620110 group. (**B**) Triglyceride levels in serum were determined after oral administration of PF04620110 for seven days. (**C**) The representative traces show the baseline renal SNA in each group on day 7. The time scale is one integrated value per 5 s. (**D**) Renal SNA was measured in each group at day 7. Renal SNA increased in fructose-fed rats. Treatment with PF04620110 did not attenuate sympathetic activation in fructose-fed rats. (**E**) NO concentrations were determined in the NTS of rats after oral administration of PF04620110 for seven days. NO concentrations decreased in fructose-fed rats. PF04620110 treatment did not improve NO concentrations in fructose-fed rats. The data are presented as the mean ± SEM (*n* = 4–8). * *p* < 0.05 versus the control group. ^#^
*p* < 0.05 versus the fructose group.

**Figure 4 nutrients-11-02581-f004:**
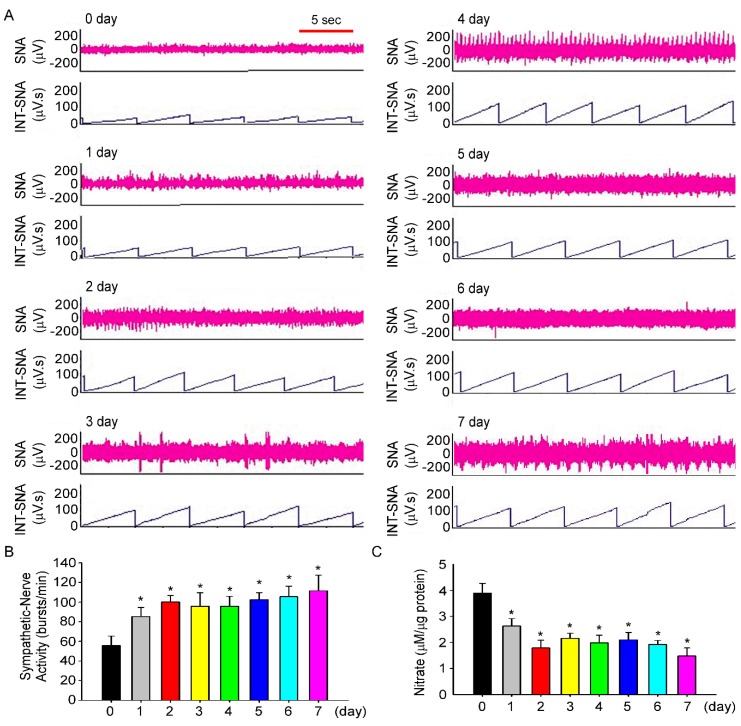
**Intake of 10% fructose water for 1 week increases SNA and decreases NO concentrations in the NTS.** (**A**) The representative traces show the baseline renal SNA in fructose-fed rats throughout the seven days of the experiment. The time scale is one integrated value per 5 s. (**B**) Renal SNA in fructose-fed rats was measured every day. (**C**) NO concentrations in the NTS of fructose-fed rats were determined throughout the 7-day research. The data are presented as the mean ± SEM (*n* = 5–7). * *p* < 0.05 versus the control group.

**Figure 5 nutrients-11-02581-f005:**
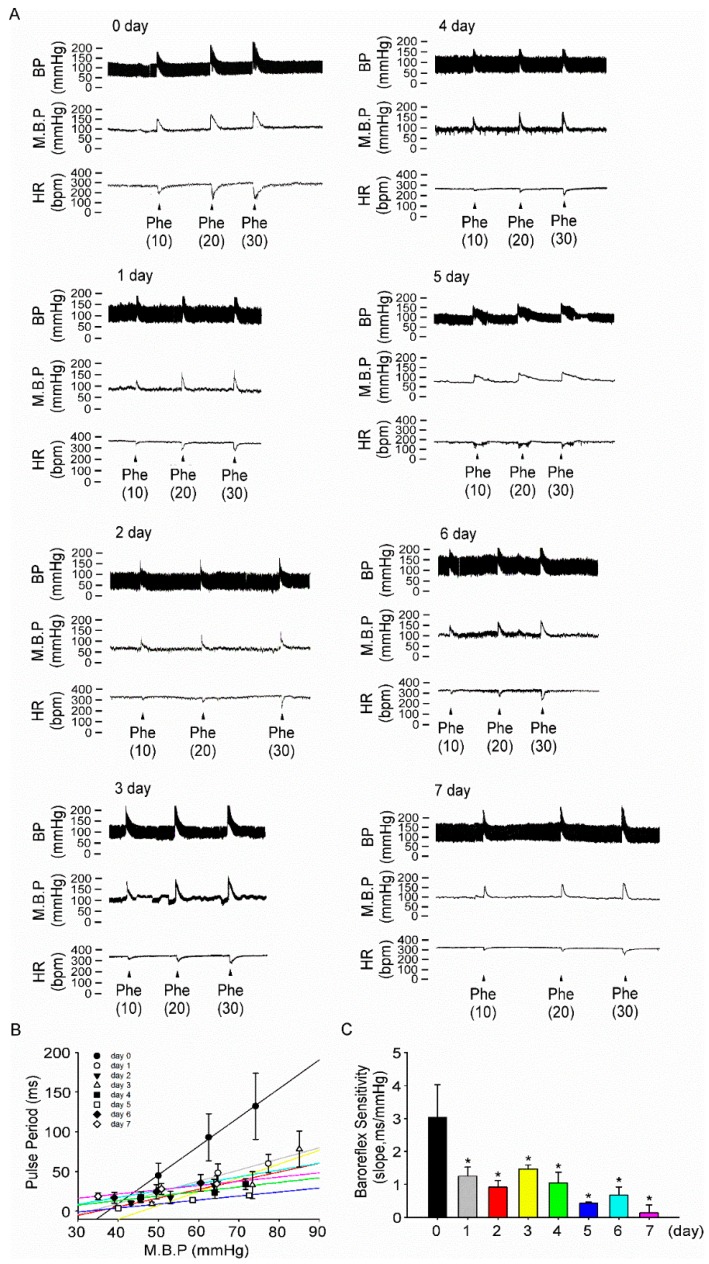
**Intake of 10% fructose water for one week impairs baroreflex response sensitivity.** (**A**) The representative traces show baroreflex responses after intravenous injection of phenylephrine (Phe: 10, 20, 30 μg/kg) in fructose-fed rats throughout the seven days of the experiment. (**B**) The points and vertical bars represent increases in the pulse period of the peak bradycardic response in response to the suppressive effects of different doses of phenylephrine. The lines connecting the points were obtained using linear regression analysis, which yielded the slopes of each group. (**C**) Effects of the NTS on baroreflex responses (slope) in the NTS to phenylephrine in fructose-fed rats throughout the seven days of the experiment. The data are presented as the mean ± SEM (*n* = 4–8). * *p* < 0.05 versus the control group.

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
