# Peer review of "Excessive Fructose Intake Impairs Baroreflex Sensitivity and Led to Elevated Blood Pressure in Rats"

_nutrients, 2019, doi:10.3390/nu11112581_

Round 1
Reviewer 1 Report
This manuscript describes a study that hypothesized that ingestion of 10% of fructose solution for 1 week will increase serum triglyceride levels which will lead to increase in renal sympathetic nerve activity. Furthermore, this will be accompanied by reduced NO in the NTS. While it is true that the mechanistic details of fructose-induced hypertension and associated cardiovascular morbidities have not been elucidated yet, this study contributes only marginally to such efforts since major flaws have been detected. These include improper use of controls and inadequately described methodology. Please find below the detailed list of suggested improvements.
Major concerns:
Line 48-50, line 52-55: these sentences were plagiarized from Ref #9.
Line 299-301: sentence plagiarized from Ref #34.
Line 64: unsupported claims have been made here. Renal afferent nerves transmit signal from the kidney to the brain (which is not made clear). Additionally, NTS is one of the sites where blood pressure integration happens (others being the SFO, PVN, RVLM – myriad of studies supporting their roles exist). Claims that afferent renal nerves rather than efferent renal nerves play a role in rat hypertension models cannot be supported without completion of experiments involving selective afferent renal nerve ablation (as shown by Banek et al.). Simply cutting the nerve while the animal is anesthetized severs both afferent AND efferent renal nerves, and the translation of such preparation to physiological phenomena is a bit of a stretch.
Line117: Blood pressure measurements are completed with the tail cuff method. While the authors provide a reference justifying the use of such methodology, this reference is 11 years old. Since then, novel, more widely accepted and more widely used method of BP measurements via telemetry should be considered or at least addressed. Major flaws of tail cuff method include the absence of measures of diastolic blood pressure, and thus mean arterial pressure. Hemodynamic perturbations can occur in either or both of the components of the blood pressure and assessing only one (i.e. systolic) without the other (i.e. diastolic) paints an incomplete picture of the status of hemodynamic function.
Line 144: While the blood pressure measurements were obtained in a conscious state, RSNA measures and baroreflex measures were obtained in an anesthetized state. The effects of anesthesia on these outcome measures should at least be addressed. Furthermore, the preparation for RSNA measurement included cutting of the renal nerve and the reason for this is unclear. More elaboration on the reason for removing the afferent input is beneficial in this study.
Line 179: The methods for measuring fructose in serum and CSF were not explained. Please describe in detail how these concentrations were obtained. Likewise, no description on the methodology of glucose concentration measurements in serum and CSF were included. Please describe. Were the concentrations of these sugars obtained under conscious or anesthetized conditions? These are important considerations since it has been shown that anesthesia raises glucose levels significantly.
Fig. 2C shows serum fructose levels expressed in mg/dL. It is disconcerting that on Day 0, there is already 20mg/dL of fructose circulating around. For a total blood volume of roughly 25mls in an 8 week old rat, this amounts to 5 mg of fructose floating in the circulatory department before they even started drinking the 10% fructose solution. Proper controls are missing in this experiment. What are the concentrations of fructose and glucose in the respective fluids on those same days in control animals drinking regular water or even 10% glucose solution (to match for caloric intake)? It is imperative that they be included to allow for proper comparisons across the time course of the experiment (i.e. days 1-7).
Line 186: “throughout the research” - please revise. How was urine output measured? How was water intake quantitated? The conventional methods to complete these experiments typically involve the use of metabolic cages. However, there is no mention of the use of metabolic cages in this study. Please explain how were water intake and urine output quantitated.
Fig. 3C: How was the quantitation of bursts completed? Conventionally, RSNA is reported as integrated value rather than the raw signal (i.e. bursts per min). In any case, detailed description of burst quantitation is missing and should be included. Additionally, it is very difficult to discern bursts in the control animal example. Also, time scale is missing on all RSNA recordings.
Figs 4 and 5: Measurements of RSNA, BRS and nitrate levels in the NTS are shown only for the experimental group, and thus proper controls are missing.
Line 249: I.V. injection of phenylephrine was given, yet no description of the procedure to instrument the rats with I.V. catheters was provided. Please provide the description of the procedure, at what time point was it done etc. It is unclear whether the data represents values obtained in the same rat across the 7 days? And if so, does that mean that the rat was anesthetized every day to obtain BRS values? What effect would every day anesthesia have on the baroreflex sensitivity?
Some groups are rather small (n=4) and it is unclear which ones were such small groups. Was the power analysis completed and if so, did it indicate adequate power with n=4?
Typos and syntax issues:
Line 59: “These are” change to “There is”
Line 62: “which is” change to “which are”
Line 62: “increased” change to “increases”
Line 70: “more easily elevated” – please revise, not scientific language
Line 82: Redundant sentence starting with ‘The aims of..”, consider deleting
Line 85: Delete “further”
Line 101: sentence structure needs revision-please correct grammar
Line 108: “biochemical levels” is unspecific and unclear, levels of what?
Line 112: sentence structure needs revision-please correct grammar
Line 206: sentence structure needs revision- please correct grammar
Line 207: “inhibited” change to “inhibiting”
Line 232: Incomplete sentence, please correct.
Line 272: “elevate” change to “elevated”
Line 282: sentence structure needs revision-please correct grammar
Line 290: Unclear sentence
Line 296: “factors” change to “factor”
Line 298: sentence structure needs revision-please correct grammar
Line 302: “did not inhibit” change to “was not inhibited” – wrong tense
Line 307: non-scientific language
Line 312-314: Unclear sentence
Author Response
Responses to comments from Reviewer 1:
We appreciate the constructive comments from Reviewer 1. We believe that the quality of the revised manuscript has been improved by incorporating the suggestions from the Reviewer. The followings are our responses to the specific issues raised:
All revisions are marked in the revised manuscript in red-color.
Major concerns:
Line 48-50, line 52-55: these sentences were plagiarized from Ref #9.
Reply: Thank you for the valuable comment. We had rewritten these sentences as follows: “High fructose consumption has shown greater adverse effects on metabolism and vascular health than did glucose consumption in several animal studies, and it causes detrimental health effects, such as fatty liver diseases [7].” (P. 2, Line 47-49) and “Fructose, a sweeter alternative to glucose, is usually used in smaller amounts than glucose. Because fructose cannot be used directly as a source of energy, the metabolisms of fructose and glucose are very different [9].” (P. 2, Line 51-53) in the Introduction section of the revised manuscript.
Line 299-301: sentence plagiarized from Ref #34.
Reply: Thank you very much for your constructive comment. We had rewritten this sentence as follows: “It has been shown that ingestion of fructose altered the secretion of hormones that regulate energy balance associated with increased SNA [34]. in the Introduction section of the revised manuscript. (P. 11, Line 293-295)
Line 64: unsupported claims have been made here. Renal afferent nerves transmit signal from the kidney to the brain (which is not made clear). Additionally, NTS is one of the sites where blood pressure integration happens (others being the SFO, PVN, RVLM – myriad of studies supporting their roles exist). Claims that afferent renal nerves rather than efferent renal nerves play a role in rat hypertension models cannot be supported without completion of experiments involving selective afferent renal nerve ablation (as shown by Banek et al.). Simply cutting the nerve while the animal is anesthetized severs both afferent AND efferent renal nerves, and the translation of such preparation to physiological phenomena is a bit of a stretch.
Reply: Thank you for your comment. The nucleus tractus solitarii (NTS) is the site where afferent fibers arising from the arterial and cardiopulmonary baroreceptors construct the first central synapses (Lo, Jan, Chiang & Tseng, 2000). A combination of functional and immunostaining techniques showed that electrical stimulation of the renal afferents increased the labelling of the neuronal activity marker Fos in many cardiovascular brain regions, such as the organum vasculosum of the lamina terminalis, the subfornical organ, the median preoptic nucleus, and the paraventricular nucleus (PVN) in the forebrain and the NTS, as well as the rostral ventrolateral medulla (RVLM) in the brainstem (Solano-Flores, Rosas-Arellano & Ciriello, 1997). Further electrophysiological evidence has shown that afferent signals from the kidney are integrated supraspinally. The effect of electrical stimulation of renal afferents on renal sympathetic nerve activity is prevented by renal denervation or spinal cord transection at the C2 level, but not by brainstem transection at the pontine–medullary junction or by lesion of the NTS (Saeki, Terui & Kumada, 1988). Renal afferent sensory nerves project to the RVLM via the NTS and PVN, where integration of afferent signals from the kidney occurs (Kumagai et al., 2012).
Line117: Blood pressure measurements are completed with the tail cuff method. While the authors provide a reference justifying the use of such methodology, this reference is 11 years old. Since then, novel, more widely accepted and more widely used method of BP measurements via telemetry should be considered or at least addressed. Major flaws of tail cuff method include the absence of measures of diastolic blood pressure, and thus mean arterial pressure. Hemodynamic perturbations can occur in either or both of the components of the blood pressure and assessing only one (i.e. systolic) without the other (i.e. diastolic) paints an incomplete picture of the status of hemodynamic function.
Reply: Thank you for m constructive suggestion. Detailed comparison of the noninvasive tail‐cuff method and the invasive telemetry blood pressure measurement techniques revealed that the tail‐cuff method systematically underestimated central blood pressure, which can be recorded simultaneously by telemetry in the same mouse; however, the tail‐cuff recordings were similar to those obtained by telemetry in undisturbed mice (Wilde et al., 2017). In fact, telemetry, the “gold‐standard” technique, is invasive and expensive; however, so far, we cannot use this technique. As the tail‐cuff method is widely used and non-invasive, we choose this method as an alternative to measure blood pressure. In our study, rat blood pressure was recorded using a CODA Non-invasive Blood Pressure System, purchased from Kent Scientific Corporation, which uses a noninvasive blood pressure measuring method. The method is also recommended by the American Heart Association in its blood pressure measuring guide for laboratory animals (Kurtz et al., 2005). The experiment consisted at least six blood pressure measurements in each laboratory animal, and data collection was conducted by the CODA system.
Line 144: While the blood pressure measurements were obtained in a conscious state, RSNA measures and baroreflex measures were obtained in an anesthetized state. The effects of anesthesia on these outcome measures should at least be addressed. Furthermore, the preparation for RSNA measurement included cutting of the renal nerve and the reason for this is unclear. More elaboration on the reason for removing the afferent input is beneficial in this study.
Reply: Thank you for your constructive suggestion. Our research has its limitations, blood pressure was measured when the rats were conscious, whereas renal SNA and baroreflex response were measured when the rats were anesthetized. However, in a recent report by Kenney and Mosher, it was determined that of the nearly 1,300 publications on studies of SNA in rats between 1981 and 2010, >1,100 were conducted in anesthetized animals. Moreover, only 10 (<1%) were conducted in conscious rats at 3 days or longer after electrode implantation (Kenney & Mosher, 2013). Therefore, we have removed redundant sentences in the Methods section of the revised manuscript, and changed the sentence as follows: “The distally cutting renal nerve was placed on a pair of silver recording electrodes and immersed in warm mineral oil.” (P. 4, Line 141-142)
Line 179: The methods for measuring fructose in serum and CSF were not explained. Please describe in detail how these concentrations were obtained. Likewise, no description on the methodology of glucose concentration measurements in serum and CSF were included. Please describe. Were the concentrations of these sugars obtained under conscious or anesthetized conditions? These are important considerations since it has been shown that anesthesia raises glucose levels significantly.
Reply: Thank you for the valuable suggestion. We obtained fasting glucose and fructose levels in serum and cerebrospinal fluid after the rats were sacrificed with excess CO2. We added a new subsection (Assay of serum biochemical parameters) and a new description in the Methods section of the revised manuscript as follows:
“Heart blood and cerebrospinal fluid (CSF) samples were collected after the rats were sacrificed with excess CO2. Fasting fructose levels in serum and CSF (EFRU-100; EnzyChrom Fructose assay kit) as well as fasting glucose levels in CSF (EBGL-100; EnzyChrom glucose assay kit) were measured using an assay kit (BioAssay System, Hayward, CA) and detected using a Biochrom Anthos Zenyth 200rt Microplate Reader (Cambridge, UK). Levels of serum total cholesterol, direct low-density lipoprotein (LDL), direct high-density lipoprotein (HDL), triglycerides, and glucose were determined using a Clinical Chemistry Analyzer (Ortho Clinical VITROS™ 350 System, Raritan, NJ, USA).” (P. 4, Line 153-159)
2C shows serum fructose levels expressed in mg/dL. It is disconcerting that on Day 0, there is already 20mg/dL of fructose circulating around. For a total blood volume of roughly 25mls in an 8 week old rat, this amounts to 5 mg of fructose floating in the circulatory department before they even started drinking the 10% fructose solution. Proper controls are missing in this experiment. What are the concentrations of fructose and glucose in the respective fluids on those same days in control animals drinking regular water or even 10% glucose solution (to match for caloric intake)? It is imperative that they be included to allow for proper comparisons across the time course of the experiment (i.e. days 1-7).
Reply: Thank you for your comment. We apologize for this mistake, and we have changed the expression of fructose levels from mg/dL to mM in Figure 2C&D of the revised manuscript. (P. 6) The mg/dL unit was obtained from the fructose conversion factor: 1 mM fructose = 18 mg/dL. Because the volume of drinking water consumed ad libitum for 1-7 days did not change much in the control rats, ad libitum intake of 10% fructose water in fructose-fed rats increased to the maximum from day 0 to day 2, and continued for 7 days (Figure 2A). Therefore, we measured glucose and fructose concentrations only in the fructose group, and not in the control rats (Figure 2 C-F).
Figure 2. Intake of 10% fructose water increases SBP and fructose concentrations in CSF and serum. (A) Daily water intake recorded for 7 days. The filled circles () represent the control group, open circles () represent the fructose group. (B) Time course of SBP examined in each group for 7 days. (C) Fructose levels in the serum of fructose-fed rats were determined for 7 days. Fructose level in the serum of fructose-fed rats significantly increased from day 2 to day 7, compared to that on day 0. (D) Fructose levels in the CSF of fructose-fed rats were determined for 7 days. The fructose level of in the CSF of fructose-fed rats significantly increased from day 1 to day 7, compared to that on day 0. (E) Glucose levels were determined in the serum of fructose-fed rats for 7 days. The glucose level in the serum of fructose-fed rats significantly increased from day 1 to day 7, compared to that on day 0. (F) Glucose levels in the CSF of the fructose-fed rats were determined for 7 days. The glucose level in the CSF of the fructose-fed rats significantly increased from day 1 to day 7, compared to that on day 0. The data are presented as the mean ± SEM (n = 4-8). *P < 0.05 versus the control group and fructose group day 0.
Line 186: “throughout the research” - please revise. How was urine output measured? How was water intake quantitated? The conventional methods to complete these experiments typically involve the use of metabolic cages. However, there is no mention of the use of metabolic cages in this study. Please explain how were water intake and urine output quantitated.
Reply: Thank you for your suggestion. We have rewritten the sentence to “we observed a continued increase in SBP during 7 days of 10% fructose water intake” in the Results sections of revised manuscript. (P. 5, Line 174-175)
We added the sentence “The rats were placed individually in metabolic cages (one per cage). Their diet and drinking water were measured, and collected 24-h urine at 10.00 a.m. each day. A measuring bottle was used to measure daily drinking water volume and urine volume.” in the Materials and Methods section of revised manuscript. (P. 3, Line 101-103).
3C: How was the quantitation of bursts completed? Conventionally, RSNA is reported as integrated value rather than the raw signal (i.e. bursts per min). In any case, detailed description of burst quantitation is missing and should be included. Additionally, it is very difficult to discern bursts in the control animal example. Also, time scale is missing on all RSNA recordings.
Reply: Thank you for your comment. The most common approach to record SNA is to apply bandpass filters with a high pass approximately 50 Hz and a low pass between 1 and 5 kHz. By calibrating the amplifier, one can calculate the microvolt level of each discharge. However, because the signal displays positive and negative voltage changes centered at approximately zero, the average level over time will be zero. To allow calculation of the overall level of SNA, either the individual spikes must be identified and counted, or, more commonly, the signal is rectified and integrated. However, SNA expressed in raw units of voltage cannot be compared between animals because the level of integrated SNA is dependent on both the frequency of burst firing and conditions at the point of contact between the electrode and nerve.
We added the time scale of RSNA recordings in Figure 3C and 4A. The time scale is one integrated value per 5 s in the Figure legend.
Figure 3C.
Figure 4A.
Figs 4 and 5: Measurements of RSNA, BRS and nitrate levels in the NTS are shown only for the experimental group, and thus proper controls are missing.
Reply: Thank you for your comment. Yes, in Figures 4 and 5, we only investigated RSNA, BRS, and nitrate levels in the NTS of fructose-fed rats because the control group showed no change from the baseline values; therefore, we did not consider including the date of the control group.
Line 249: I.V. injection of phenylephrine was given, yet no description of the procedure to instrument the rats with I.V. catheters was provided. Please provide the description of the procedure, at what time point was it done etc. It is unclear whether the data represents values obtained in the same rat across the 7 days? And if so, does that mean that the rat was anesthetized every day to obtain BRS values? What effect would every day anesthesia have on the baroreflex sensitivity?
Reply: Thank you for your suggestion. We added these sentences as follows: “Under anesthesia with Zoletil50 (50 mg/kg intraperitoneal, and 10 mg/kg intravenous if necessary) in rats, a polyethylene cannula (PE 50) was inserted into their abdominal aorta through the right femoral artery to measure the mean arterial pressure and heart rate. A second cannula was inserted into the femoral vein for phenylephrine administration.” to replace “The changes of arterial pressure and heart rate were measured intra-arterially in anesthetized rats.” in the Methods section of the revised manuscript. (P.4, Line 146-149)
Each data represent an individual rat, with no repeated anesthesia and adaptive problems. The parameters were completed measured and the rats were sacrificed. Anesthesia of the same rat every day may cause adaptability and lead to incorrect data.
Some groups are rather small (n=4) and it is unclear which ones were such small groups. Was the power analysis completed and if so, did it indicate adequate power with n=4?
Reply: Thank you for your comment. We followed the rules of the animal protection law and considered the 3Rs (reduction, refinement, and replacement) to carry out the experiment. However, our results were almost always obtained from different rats. We did not repeat all experiments, except for the drinking water, diet, and urine volume analyses.
We used the nonparametric method for data analysis. All data are expressed as the means ± SEM. Mann-Whitney U-test (for control and fed 10% fructose groups comparisons) or Kruskal-Wallis followed by one-way analysis of variance was applied to compare differences between groups. A value of P < 0.05 was considered significant.
Typos and syntax issues:
Reply: Thank you for the valuable suggestions. We have revised the grammar, typographical errors, and language-related errors throughout the revised manuscript.
Line 59: “These are” change to “There is”
Reply: We have “These are” changed to “There is” in the revised manuscript. (P.2, Line 57)
Line 62: “which is” change to “which are”
Reply: We have “which is” changed to “these parameters are” in the revised manuscript. (P.2, Line 60)
Line 62: “increased” change to “increases”
Reply: We have “increased” changed to “increases” in the revised manuscript. (P.2, Line 60)
Line 70: “more easily elevated” – please revise, not scientific language
Reply: We have revised the sentence to “In a previous study, compared with those in glucose-fed rats, serum triglyceride levels increased to a greater degree in fructose-fed rats” in the revised manuscript. (P.2, Line 68-69)
Line 82: Redundant sentence starting with ‘The aims of..”, consider deleting
Reply: We have deleted “The aims of” in the revised manuscript. (P.2, Line 80)
Line 85: Delete “further”
Reply: We have deleted “further” in the revised manuscript. (P.2, Line 83)
Line 101: sentence structure needs revision-please correct grammar
Reply: We have revised the sentence to “fructose group, WKY rats were provided with 10% fructose water ad libitum for 0 to 7 days” in the revised manuscript. (P.3, Line 97-98)
Line 108: “biochemical levels” is unspecific and unclear, levels of what?
Reply: We have revised the sentence to “the levels of serum biochemical parameters” in the revised manuscript. (P.3, Line 107)
Line 112: sentence structure needs revision-please correct grammar
Reply: We have revised the sentence “To confirm whether fructose-induced hypertension was inhibited by triglyceride inhibition, rats were divided into four groups in this one-week experiment:” in the revised manuscript. (P.3, Line 110-112)
Line 206: sentence structure needs revision- please correct grammar
Reply: We have revised the sentence “If the BP of rats increased after consumption of 10% fructose water, …” in the revised manuscript. (P.6, Line 202)
Line 207: “inhibited” change to “inhibiting”
Reply: We have “inhibited” changed to “inhibition” in the revised manuscript. (P.6, Line 202)
Line 232: Incomplete sentence, please correct.
Reply: We have revised the sentence “The next experiment determined whether intake of 10% fructose water for 1 week increases SBP and BP, we…” in the revised manuscript. (P.8, Line 227-238)
Line 272: “elevate” change to “elevated”
Reply: We have “elevate” changed to “elevated” in the revised manuscript. (P.11, Line 267)
Line 282: sentence structure needs revision-please correct grammar
Reply: We have revised the sentence “increased fructose intake for 0 to 2 days evoked increases in BP levels, and the high BP levels were maintained until day 7” in the revised manuscript. (P.11, Line 277-278)
Line 290: Unclear sentence
Reply: We have rewritten the sentence to “Hypertension inhibition by prazosin did not reduce level of triglyceride in serum.” in the revised manuscript. (P.11, Line 284-285)
Line 296: “factors” change to “factor”
Reply: We have changed “factors” to “factor” in the revised manuscript. (P.11, Line 291)
Line 298: sentence structure needs revision-please correct grammar
Reply: We have revised the sentence to “elevated BP was increased by sympathoexcitation in 20% fructose-induced, salt-sensitive, hypertensive rats” in the revised manuscript. (P.11, Line 292-293)
Line 302: “did not inhibit” change to “was not inhibited” – wrong tense
Reply: We have changed “did not inhibit” to “was not inhibited” in the revised manuscript. (P.11, Line 296)
Line 307: non-scientific language
Reply: We have rewritten the sentence to “With regarding to the relationship between baroreflex sensitivity and NO level in the NTS, …” in the revised manuscript. (P.11, Line 301)
Line 312-314: Unclear sentence
Reply: We have rewritten the sentence to “In terms of the mechanisms by which fructose in the CSF induces suppression of NO levels in the NTS and further damages baroreflex sensitivity, …” in the revised manuscript. (P.11, Line 306-308)
Reference:
Kenney MJ, & Mosher LJ (2013). Translational physiology and SND recordings in humans and rats: a glimpse of the recent past with an eye on the future. Auton Neurosci 176: 5-10.
Kumagai H, Oshima N, Matsuura T, Iigaya K, Imai M, Onimaru H, et al. (2012). Importance of rostral ventrolateral medulla neurons in determining efferent sympathetic nerve activity and blood pressure. Hypertens Res 35: 132-141.
Kurtz TW, Griffin KA, Bidani AK, Davisson RL, Hall JE, Subcommittee of P, et al. (2005). Recommendations for blood pressure measurement in humans and experimental animals. Part 2: Blood pressure measurement in experimental animals: a statement for professionals from the subcommittee of professional and public education of the American Heart Association council on high blood pressure research. Hypertension 45: 299-310.
Lo WC, Jan CR, Chiang HT, & Tseng CJ (2000). Modulatory effects of carbon monoxide on baroreflex activation in nucleus tractus solitarii of rats. Hypertension 35: 1253-1257.
Saeki Y, Terui N, & Kumada M (1988). Physiological characterization of the renal-sympathetic reflex in rabbits. Jpn J Physiol 38: 251-266.
Solano-Flores LP, Rosas-Arellano MP, & Ciriello J (1997). Fos induction in central structures after afferent renal nerve stimulation. Brain Res 753: 102-119.
Wilde E, Aubdool AA, Thakore P, Baldissera L, Jr., Alawi KM, Keeble J, et al. (2017). Tail-Cuff Technique and Its Influence on Central Blood Pressure in the Mouse. J Am Heart Assoc 6.

Reviewer 2 Report
General Comment: The study investigated the effects of excessive fructose intake on blood pressure, triglyceride synthesis, sympathetic nerve activity, baroreflex sensitivity, and nitric oxide in the NTS in rats. Results clearly showed that fructose intake increased blood pressure, reduced baroreflex sensitivity, and decreased NO in the NTS. These findings are important in light of the increased fructose consumption and the high prevalence of hypertension in the global society.
I suggest specifying the number of rats in each group be included in the Methods.
Systolic blood pressure measurement via the CODA 8 device is indeed a valid method. I suggest a little more information explaining the measurement would be helpful. The technique measures BP for 10 to 15 times per cycle. How many valid measurements during each cycle were considered to obtain the SBP for a rat on a given day? Was blood pressure measured during the same time of day for each day?
Was food intake measured? In line 185 “….the body weight and the diet maintained the same”. Does that mean food intake was equal between the two groups? Food intake of rodents consuming fructose water should be significantly different from the normal group.
The results on triglycerides are interesting. Results in the study support the finding that the fructose-induced increase in renal SNA as being involved in the fructose-induced increase in SBP. Even though the effect of PF04620110 on SBP was small (Figure 3A), this study also supports the position that fructose consumption increases blood pressure via multiple effects. I suggest that be acknowledged in the Discussion.
Increased triglyceride levels indeed associates with metabolic syndrome and hypertension. A statement on the mechanism of triglyceride-induced increase in blood pressure would be helpful. Was there any evidence that triglycerides affect sympathetic nervous system?
Author Response
Responses to comments from Reviewer 2:
We appreciate the constructive comments of Reviewer 2. We believe that the quality of the revised manuscript has been improved by incorporating the suggestions from the Reviewer. The followings are our responses to the specific issues raised:
All the revisions are marked in the revised manuscript in red-color.
I suggest specifying the number of rats in each group be included in the Methods.
Reply: Thank you for your constructive suggestion. We added the number of rats in each group in the Methods of the revised manuscript. (P. 3)
Systolic blood pressure measurement via the CODA 8 device is indeed a valid method. I suggest a little more information explaining the measurement would be helpful. The technique measures BP for 10 to 15 times per cycle. How many valid measurements during each cycle were considered to obtain the SBP for a rat on a given day? Was blood pressure measured during the same time of day for each day?
Reply: Thank you for your valuable suggestion. We measured blood pressure in each rat during the same time of day before noon.
Each recording session consisted of 20 to 30 inflation and deflation cycles per set, of which the first 5 cycles were “acclimation” cycles and not used in the analysis, whereas the subsequent cycles were used. 20 to 30 individual readings were obtained in rapid succession during the measurement. The highest and lowest readings were eliminated, and 10 to 20 remaining readings were averaged. We added the description in the Materials and Methods section of the revised manuscript. (P. 4, Line 125-129)
Was food intake measured? In line 185 “….the body weight and the diet maintained the same”. Does that mean food intake was equal between the two groups? Food intake of rodents consuming fructose water should be significantly different from the normal group.
Reply: Thank you for your comment. The rats were placed individually in metabolic cages (one per cage). Their diet and drinking water were measured, and collected 24-h urine at 10.00 a.m. each day. A measuring bottle was used to measure daily drinking water volume and urine volume. We added the description In the Materials and Methods section of the revised manuscript. (P. 3, Line 100-102).
Body weight and diet did not change much during the 7-day experiment, but drinking water and urine volumes were significantly different between the fructose and normal groups. We revised the sentence to “body weight and diet did not change much” in the Results section of the revised manuscript. (P. 5, Line 182)
The results on triglycerides are interesting. Results in the study support the finding that the fructose-induced increase in renal SNA as being involved in the fructose-induced increase in SBP. Even though the effect of PF04620110 on SBP was small (Figure 3A), this study also supports the position that fructose consumption increases blood pressure via multiple effects. I suggest that be acknowledged in the Discussion.
Reply: Thank you for your suggestion. Increasing number of studies has proven that medicines with multiple effects, such as statins, can lower blood fat and blood pressure. Statins not only lower low-density lipoprotein cholesterol level but also inhibit many of the structural and functional components of the arteriosclerotic process. In fact, 14 studies have proved that statin induced a decrease in blood pressure, but 11 studies showed no effect (Mangat, Agarwal & Rosendorff, 2007). Despite predictions made on the basis of the vasoprotective actions of statins, their blood-pressure-lowering effects are, at best, modest. DGAT1 inhibitors (such as PF04620110) are another class of inhibitors that may be used in addition to statins; they inhibit triacylglycerol synthesis in cells and rodents. In addition, our previous studies indicated that superoxide production increases in the NTS of rats fed fructose for one week (Cheng et al., 2014). Fructose-induced neurogenic hypertension might be mediated by the activation of p38, followed by phosphorylation of the insulin receptor substrate 1 ser307, which might occur via superoxide overexpression in the NTS (Cheng et al., 2017). In this study, we showed that fructose intake may reduce NO levels in the NTS and cause baroreflex dysfunction, which further stimulates SNA and induces the development of high BP. The above description was added in the Discussion section of the revised manuscript. (P. 11-12, Line 308-313)
Increased triglyceride levels indeed associates with metabolic syndrome and hypertension. A statement on the mechanism of triglyceride-induced increase in blood pressure would be helpful. Was there any evidence that triglycerides affect sympathetic nervous system?
Reply: Thank you for your comment. There is a clear evidence on the role of the sympathetic nervous system in regulating hepatic lipogenesis and VLDL-triglyceride production (Carreno & Seelaender, 2004), in which increased sympathetic outflow to the liver generally leads to an increase in VLDL-triglyceride production, resulting in increased availability of free fatty acids to be used by peripheral organs (i.e., white adipose tissue, brown adipose tissue, and muscle tissue) (Bruinstroop et al., 2012). However, our data showed that the DGAT1 inhibitor used for inhibiting serum triglyceride led to a slight decrease in blood pressure, but did not affect sympathetic nerve activity. In the future, we will conduct more experiments to elucidate how triglycerides affect sympathetic nerve activity.
Reference:
Bruinstroop E, Pei L, Ackermans MT, Foppen E, Borgers AJ, Kwakkel J, et al. (2012). Hypothalamic neuropeptide Y (NPY) controls hepatic VLDL-triglyceride secretion in rats via the sympathetic nervous system. Diabetes 61: 1043-1050.
Carreno FR, & Seelaender MC (2004). Liver denervation affects hepatocyte mitochondrial fatty acid transport capacity. Cell Biochem Funct 22: 9-17.
Cheng PW, Ho WY, Su YT, Lu PJ, Chen BZ, Cheng WH, et al. (2014). Resveratrol decreases fructose-induced oxidative stress, mediated by NADPH oxidase via an AMPK-dependent mechanism. Br J Pharmacol 171: 2739-2750.
Cheng PW, Lin YT, Ho WY, Lu PJ, Chen HH, Lai CC, et al. (2017). Fructose induced neurogenic hypertension mediated by overactivation of p38 MAPK to impair insulin signaling transduction caused central insulin resistance. Free Radic Biol Med 112: 298-307.
Mangat S, Agarwal S, & Rosendorff C (2007). Do statins lower blood pressure? J Cardiovasc Pharmacol Ther 12: 112-123.

Round 2
Reviewer 1 Report
The original comment on the blood pressure measurements being obtained in a conscious state and RSNA measures and baroreflex measures being obtained in an anesthetized state may have been misunderstood. Can you please comment (in the discussion section) on the anticipated impact of anesthesia on RSNA and baroreflex sensitivity vs. conscious BP measurements? Also can you please comment (again, in the discussion section) on why you cut the renal nerve (i.e. why you removed afferent input) for RSNA measurements? The original comment on fructose measurements may also have been misunderstood. Expressing fructose in mg/dL was not the intended criticism, it is totally fine to use these units. The point of this comment was that on day 0, there is already quite a bit of fructose in the blood before the rats even start drinking it. Is the same observed for rats on normal water? Does their blood fructose fluctuate throughout the 7 days? The answers to these questions can only be demonstrated by showing results for the control group, and allowing the reader to compare the fructose and control groups. The results on blood fructose concentration for the control group are still missing from the manuscript and should be included. Please indicate the make (i.e. the manufacturer) of the metabolic cagesFor results in Figs 4 and 5: Measurements of RSNA, BRS and nitrate levels in the NTS are shown only for the experimental group, and thus proper controls are missing. The data for the control groups should still be shown such that the reader can draw conclusions whether there are differences between control and experimental groups. If the available space is the limitation, then the results from the control group could be reported in the text (with values). In other words, the results from the control group should be reported in any way and for every experiment as part of a rigorous scientific design.
It is always commendable to learn that investigators are following the 3Rs principle. However, unless you have completed power analysis showing that n=4 is enough for effect size, then the study may be inadequately powered. The described statistics are fine, but you are still missing the power analysis.
Author Response
Responses to comments from Reviewer 1:
We appreciate the constructive comments from Reviewer 1. We believe that the quality of the revised manuscript has been improved by incorporating the suggestions from the Reviewer. The following are our responses to the specific issues raised:
All revisions are marked in the revised manuscript in red.
The original comment on the blood pressure measurements being obtained in a conscious state and RSNA measures and baroreflex measures being obtained in an anesthetized state may have been misunderstood. Can you please comment (in the discussion section) on the anticipated impact of anesthesia on RSNA and baroreflex sensitivity vs. conscious BP measurements?
Reply: Thank you for these constructive suggestions. Generally, anesthetics affect the basal levels of sympathetic nerve activity to regulate blood pressure, which plays an important role in controlling cardiovascular function [36,37]. Sun et al. indicated that anesthesia attenuated the excitatory response of RSNA and HR to anaphylactic hypotension, while these excitatory responses were attenuated by anesthetics in the order ketamine-xylazine > urethane = pentobarbital [38]. Bencze et al. found that pentobarbital anesthesia had a modest influence on the BP level and its maintenance by the above vasoactive systems [37]. In addition, anesthetics may inhibit the primary area in the baroreceptor reflex pathway of the central nervous system, resulting in attenuation of the baroreceptor reflex [39]. Therefore, sympathetic nerve activity and cardiovascular function may be affected by anesthetic. However, there were some limitations to our study. First, we measured BP during consciousness, whereas RSNA and baroreflex sensitivity measurements were performed under the anesthetic. Second, our sample size was relatively small and the study used nonparametric methods for analysis. Third, the fructose group was used as the control group only on day 0. We added a description of these points to the Discussion section of the revised manuscript. (P. 11, Line 306–318)
Also can you please comment (again, in the discussion section) on why you cut the renal nerve (i.e. why you removed afferent input) for RSNA measurements?
Reply: Thank you for your comment. We measured RSNA by multifiber recording directly from nerves innervating the kidneys. We cut the renal nerve and placed them on a pair of silver recording electrodes (record only one-way signals to avoid two-way signal interference) and then immersed the sample in warm mineral oil (for insulation). We have rewritten the description in the Materials and Methods section of the revised manuscript. (P. 4, Line 143–144)
The original comment on fructose measurements may also have been misunderstood. Expressing fructose in mg/dL was not the intended criticism, it is totally fine to use these units. The point of this comment was that on day 0, there is already quite a bit of fructose in the blood before the rats even start drinking it. Is the same observed for rats on normal water? Does their blood fructose fluctuate throughout the 7 days? The answers to these questions can only be demonstrated by showing results for the control group, and allowing the reader to compare the fructose and control groups. The results on blood fructose concentration for the control group are still missing from the manuscript and should be included.
Reply: We apologize for this unclear information and we appreciate your understanding. First, we combined the control groups, repeated the remeasurements, and found that the previous values were indeed too high. Second, we added the control group to the subsequent experiments. However, because of the time relationship, we only added control groups on days 0, 3, and 7. We hope that the reviewer can understand this point. We observed the same results on different days in control rats that were provided drinking water. Serum fructose levels were higher compared to those in the control group after drinking 10% fructose water. We have added the remeasurement results to the revised manuscript (Figure 2C). (P. 6)
Please indicate the make (i.e. the manufacturer) of the metabolic cages.
Reply: Thank you for the suggestion. We used Techniplast metabolic cages for rats (Braintree Scientific, Inc., Braintree, MA, USA). We have added a description of the manufacturer to the Materials and Methods section of the revised manuscript. (P. 3, Line 101)
For results in Figs 4 and 5: Measurements of RSNA, BRS and nitrate levels in the NTS are shown only for the experimental group, and thus proper controls are missing. The data for the control groups should still be shown such that the reader can draw conclusions whether there are differences between control and experimental groups. If the available space is the limitation, then the results from the control group could be reported in the text (with values). In other words, the results from the control group should be reported in any way and for every experiment as part of a rigorous scientific design.
Reply: Thank you for the valuable suggestion. We have added the control group to the subsequent experiments (RSNA, BRS, and nitrate levels in the NTS). However, because of the time relationship, we only added control groups on days 0, 3, and 7. We observed the same results (RSNA, BRS, and nitrate levels in the NTS) on different days in the control rats that were provided drinking water. RSNA, BRS, and nitrate levels in the NTS were higher compared to those in the control group after drinking 10% fructose water. These results were uploaded in the supplement files.
Supplemental Figure 1. Intake of 10% fructose water for 1 week increases SNA and decreases NO concentrations in the NTS. (A) Representative traces show the baseline renal SNA in fructose-fed rats throughout the 7 days of experiment. The time scale is one integrated value per 5 s. (B) Renal SNA in fructose-fed rats was measured on days 0, 3, and 7. (C) NO concentrations in the NTS of fructose-fed rats were determined throughout the 7-day research. The data are presented as the mean ± SEM (n = 5–7). *P < 0.05 versus the control group.
Supplemental Figure 2. Intake of 10% fructose water for 1 week impairs baroreflex response sensitivity. (A) Representative traces show baroreflex responses after intravenous injection of phenylephrine (Phe: 10, 20, 30 mg/kg) in fructose-fed rats on days 0, 3, and 7 of the experiment. (B) The points and vertical bars represent increases in the pulse period of the peak bradycardic response in response to the suppressive effects of different doses of phenylephrine. The lines connecting the points were obtained by linear regression analysis, which yielded the slopes of each group. (C) Effects of the NTS on baroreflex responses (slope) in fructose-fed rats on days 0, 3, and 7 of the experiment. The data are presented as the mean ± SEM (n = 4–6). *P < 0.05 versus the control group.
It is always commendable to learn that investigators are following the 3Rs principle. However, unless you have completed power analysis showing that n=4 is enough for effect size, then the study may be inadequately powered. The described statistics are fine, but you are still missing the power analysis.
Reply: We thank the reviewer for bringing our attention to this point. A post hoc power analysis using G-Power 3.1.9.2 software was used to determine the power of the present study. Based on the results of all other variables, effect sizes from 0.2 to 14.6 were calculated. Using these effect sizes, an alpha of 0.05, and sample size from 4 to 8, the power ranging from 0.06 to 0.99 was calculated. We have added this description to the Materials and Methods section of the revised manuscript. (P. 4, Line 166–169)
References:
Matsukawa, K.; Ninomiya, I. Anesthetic effects on tonic and reflex renal sympathetic nerve activity in awake cats. Am J Physiol 1989, 256, R371-378. Bencze, M.; Behuliak, M.; Zicha, J. The impact of four different classes of anesthetics on the mechanisms of blood pressure regulation in normotensive and spontaneously hypertensive rats. Physiol Res 2013, 62, 471-478. Staff, P.O. Correction: Effects of anesthetics on the renal sympathetic response to anaphylactic hypotension in rats. PLoS One 2015, 10, e0118042. Fisher, M.T.; Fisher, J.L. Activation of alpha6-containing gabaa receptors by pentobarbital occurs through a different mechanism than activation by gaba. Neurosci Lett 2010, 471, 195-199.
Reviewer 2 Report
All responses to my concerns are thorough and satisfactory. Thank you
Author Response
Comments and Suggestions for Authors
All responses to my concerns are thorough and satisfactory. Thank you.
Reply: We thank you for considering our manuscript for publication in Nutrients.

Round 3
Reviewer 1 Report
Thank you for including the data from control animals. It is perfectly acceptable to show days 0,3, and 7 for controls in the context of your study.